# Autoimmunity and Immunodeficiency in Severe SARS-CoV-2 Infection and Prolonged COVID-19

Jenny Valentina Garmendia [1], Alexis Hipólito García [2], Claudia Valentina De Sanctis [1], Marián Hajdúch [1,3] and Juan Bautista De Sanctis [1,3,*]

1   Institute of Molecular and Translational Medicine, Faculty of Medicine and Dentistry, Palacky University, 779 00 Olomouc, Czech Republic
2   Institute of Immunology, Faculty of Medicine, Universidad Central de Venezuela, Caracas 1040, Venezuela
3   Czech Institute of Advanced Technology in Research [Catrin], Palacky University, 779 00 Olomouc, Czech Republic
*   Correspondence: juanbautista.desanctis@upol.cz

**Abstract:** SARS-CoV-2 causes the complex and heterogeneous illness known as COVID-19. The disease primarily affects the respiratory system but can quickly become systemic, harming multiple organs and leading to long-lasting sequelae in some patients. Most infected individuals are asymptomatic or present mild symptoms. Antibodies, complement, and immune cells can efficiently eliminate the virus. However, 20% of individuals develop severe respiratory illness and multiple organ failure. Virus replication has been described in several organs in patients who died from COVID-19, suggesting a compromised immune response. Immunodeficiency and autoimmunity are responsible for this impairment and facilitate viral escape. Mutations in IFN signal transduction and T cell activation are responsible for the inadequate response in young individuals. Autoantibodies are accountable for secondary immunodeficiency in patients with severe infection or prolonged COVID-19. Antibodies against cytokines (interferons α, γ and ω, IL1β, IL6, IL10, IL-17, IL21), chemokines, complement, nuclear proteins and DNA, anticardiolipin, and several extracellular proteins have been reported. The type and titer of autoantibodies depend on age and gender. Organ-specific autoantibodies have been described in prolonged COVID-19. Their role in the disease is under study. Autoimmunity and immunodeficiency should be screened as risk factors for severe or prolonged COVID-19.

**Keywords:** SARS-CoV-2; COVID-19; prolonged COVID-19; autoimmunity; autoantibodies; primary immune; secondary immunodeficiency

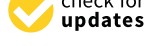



## 1. Introduction

Coronavirus disease 19 (COVID-19) is a complex and heterogeneous medical entity. It is caused by SARS-CoV-2 (severe acute respiratory syndrome coronavirus 2). SARS-CoV-2 is a β coronavirus with non-segmented positive RNA and a characteristic protein in the capsid forming a crown [1,2]. It is homologous to SARS-CoV and MERS-CoV (Middle-East respiratory syndrome) [1,2]. The virus is transmitted by aerosols or aerosolized secretions, sneezing, and coughing. The R0, the average number of infected individuals arising by the transmission from one infected individual, is 3–5, yet the most recent variant, omicron, displays a very high rate of infection and re-infection never seen before with previous strains [α, β, γ, δ] with an R0 of ~10 [3–10]. Although most infections, around 80%, are mild or asymptomatic, 20% result in severe or fatal illnesses. The severity of the symptoms can provoke a long-lasting form of the disease (prolonged COVID-19) [11,12].

SARS-CoV-2 is a pleomorphic virus with four structural proteins, spike (S), envelope (E), membrane (M), and nucleocapsid (N). (1) The S protein is highly glycated, and it is the primary but not the only protein that mediates the binding of the virus to the cell [13]. Angiotensin-2 converting enzyme receptor, expressed predominantly in epithelial cells

type II, and to a lesser extent in club cells, is one of the primary receptors [13]. However, the S protein binds to other receptors, CD26 (MERS virus receptor), CD147, neuropilin 1 (NRP-(1)) and several other receptors in different cell types [13]. (2) The E protein is a transmembrane protein that promotes viral folding and excretion and activates ion channel activity in target cells [14]. (3) The M protein, a dimer responsible for maintaining the virion's structural form, inhibits interferons type I and III signaling [15]. (4) The N protein, bound to the RNA, antagonizes interferon signaling [16,17]. The virus contains 16 non-structural proteins (NSP) and 14 open-reading frame (ORF) proteins. NSP3 is an essential cysteine protease whose role is to degrade the viral protein pp1a to form NSP1, NSP2 and NSP3 [17]. The proteins NSP1, NSP2, NSP6, NSP8, NSP12, NSP13, NSP14, NSP15, ORF3a, ORF6, ORF8, ORF9b and ORF10 have been shown to interfere with immune cell responses and interferon function, facilitating virus escape [17–22]. The M protein has also been shown to inhibit NFkB activation [15,21]. SARS-CoV-2 shares miRNAs with its host (8066, 5197, 3611, 3934-3p, 1307-3p, 3691-3p, 1468-5p), and blocks other important cell mRNAs, leading to immune evasion [23–25].

COVID-19 is mainly known to cause severe acute respiratory symptoms (rhinorrhea, cough, nasal obstruction, breathing difficulties and adult-onset respiratory distress) [26–29]. However, the virus can affect several organs. The other relevant affections of the infection are: (1) the gastrointestinal system (diarrhea, nausea and vomiting); (2) the cardiovascular system (myocarditis, pericarditis, myocardial infarct, venous and arterial thrombosis); (3) the central nervous system (encephalitis, compromised sense of smell and taste) [26–29]; (4) endothelial dysfunction and damage leading to respiratory insufficiency and multiple organ failure [28]. Several unspecific manifestations of COVID-19 can be easily confused with symptoms provoked by other ailments. Fever, myalgia, throat soreness, multiple skin rashes, arthritis, serositis, ocular and cerebral alterations, endocrine dysfunction, kidney failure, diabetes, susceptibility to opportunistic infections, and others have been described [26–29]. However, in a group of patients, a rare but severe disorder associated with COVID-19 termed multisystem inflammatory syndrome (MIS) can affect children (MIS-C) and adults (MIS-A) [30–34]. It involves different body parts, the heart, lungs, kidneys, brain, skin, eyes, or gastrointestinal organs, which become inflamed, and has been extensively reviewed by Hoste and coworkers [31] and Hossein and coworkers [32]. In a meta-analysis, Kunar and coworkers [32] reported that MIS-A is observed predominantly in males; the most affected tissues are cardiovascular, gastrointestinal and mucocutaneous. Fever and skin rash are frequent in MIS-A. In children, MIS can be severe; however, the mortality is low compared to adults [31]. Male children are more affected by MIS-C, and obesity seems to be the main comorbidity [31]. Extensive guidelines of MIS-C from the American College of Rheumatology have emphasized the follow-up of patients with MIS and the screening of autoantibodies [33]. Vaccination was also shown to protect against MIS-C [34]. The recovery of children differs from adults, and the sequela of COVID-19 is more common in adults than children [30–34].

The individuals with more risk of having severe COVID-19 are: (1) elderly patients with one or more comorbidities (obesity, diabetes, hypertension, chronic obstructive pulmonary disease, cardiovascular and/or kidney disease and cancer); (2) individuals with primary or secondary immunodeficiencies; (3) the immunosuppressed; (4) organ or tissue transplanted patients; and (5) pregnant women [28,29,33,35]. SARS-CoV-2 replication generating new variants of concern in immunodeficient patients has been challenging during the pandemic [36,37].

In a previous article, we reviewed, in detail, the immune response against SARS-CoV-2 viral infection [38]. However, new elements in the immune response to severe and prolonged COVID-19 are analyzed in the present review.

## 2. Brief Overview of the Immune Response against SARS-CoV-2

The abrogation and elimination of SARS-CoV-2 require both effective innate and adaptative immune responses. An innate immune response mediated by toll-like receptors

(TLRs) and pattern recognition receptors (PRRs), interferon production, complement activation, macrophages and NK cells; and an adaptive response in which T cells act against proteins expressed by SARS-CoV-2, and B cells produce and release neutralizing antibodies against the virus [38–40]. Vaccines activate T cell responses, enhancing antibody production and eliminating virus-infected cells.

Antigen-presenting cells (APCs) can phagocyte or endocyte viral antigens similar to what was described after infection with SARS-CoV-1 [38,41]. Interferon production increases the antiviral activity of T cells, but this response can be suppressed in patients with severe COVID-19 [41,42]. SARS-CoV-2, like other RNA viruses, induces interferons type I and type III production by activating different receptors (cytosolic retinoid acid-inducible gene I, melanoma differentiation-associated protein-5, and TLR-3) [11,38,42,43]. IFNs type I and type III activate multiple antiviral genes through their specific receptors to eliminate the virus, induce tissue repair and promote an adaptive immune response [44–46]. On the other hand, IFN I could exacerbate inflammation during severe COVID-19 [46,47]. Critically ill COVID-19 patients have decreased or nonexistent IFN I activity in blood and nasal epithelium due to the lack of protein or the inhibition of the signal pathway indicated by the cytokine [48,49]. The absence of an IFN biological effect can be due to viral proteins of miRNA impairing transcription, expression of the cytokine, or blocking signal transduction. In addition, inadequate or autoimmune responses can also have the same effect.

Activated T lymphocytes migrate to tissues to eliminate infected cells. However, cell death and viral proteins activate the inflammasome to produce an excessive number of cytokines (cytokine storm) IL1β, TNFα, IL-6, IL-18 and chemokines [3,38,50–53]. These proinflammatory cytokines increase capillary permeability and cellular adhesion and recruit monocytes and neutrophils. In the most severe forms of COVID-19, IL-5, IL-13, IgE, and eosinophils can be at high levels [3,38,50–53]. Recruited neutrophils phagocyte the virus and cell debris, but these cells also produce oxygen and nitrogen radicals, which may injure the neighboring tissues [3,38,50–53]. Damaged white blood cells and endothelial cells also secrete proinflammatory mediators (arachidonic acid, prostaglandins and leukotrienes) [3,38,50–53]. Therefore, it is crucial to administer dexamethasone during the early stages of COVID-19 or as soon as possible to avoid severe illness and other complications [54,55].

Generally, the immune system can efficiently control viral infection [50]. As expected, vaccination has decreased the number of individuals with severe COVID-19 [38]; however, several issues still require investigation.

In patients with an impaired immune response, viral replication may continue in several organs, increasing viral load and protein [36]. The infection of other organs that do not express ACE2 receptors has led to new research on how the virus can infect the cells [13]. Multiple cell receptors binding to the S protein, and several cell receptors or coreceptors bind to other viral proteins enhancing viral infection. In addition, the viral entry has been observed by transcytosis, described primarily in intestinal cells [56,57], and the antibody virus complex binding to immunoglobulin Fc receptors [56–58]. The general illustration of the different virus-cell interactions is shown in Figure 1.

The endocytosis of the virus, independent of the mechanism, induces the expression of heat shock proteins, which may be responsible for the presentation of viral and cellular proteins [59,60]. This presentation may induce polyclonal immune cell activation and the production of antibodies against cellular antigens (autoantibodies) [61].

ACE2, CD26, CD147, and neuropilin 1 (NRP-1) are the primary receptors interacting with the S protein [13]. However, other receptors have been shown as possible binding sites [13]. The variants and subvariants of SARS-CoV-2 can bind with different potency to the receptors and may affect several cellular processes [13]. CD26 and CD147 are mainly expressed in leukocytes, particularly in Th1 cells and macrophages [13,62]. The infection of both cell types may be crucial in the outcome of the immune response against the virus and may explain the lymphopenia described in severe COVID-19 [61]. Shen and coworkers [63]

have reported the infection of T lymphocytes and the hyperactivation of T cells, which may be responsible for the decrease in T cells in COVID-19.

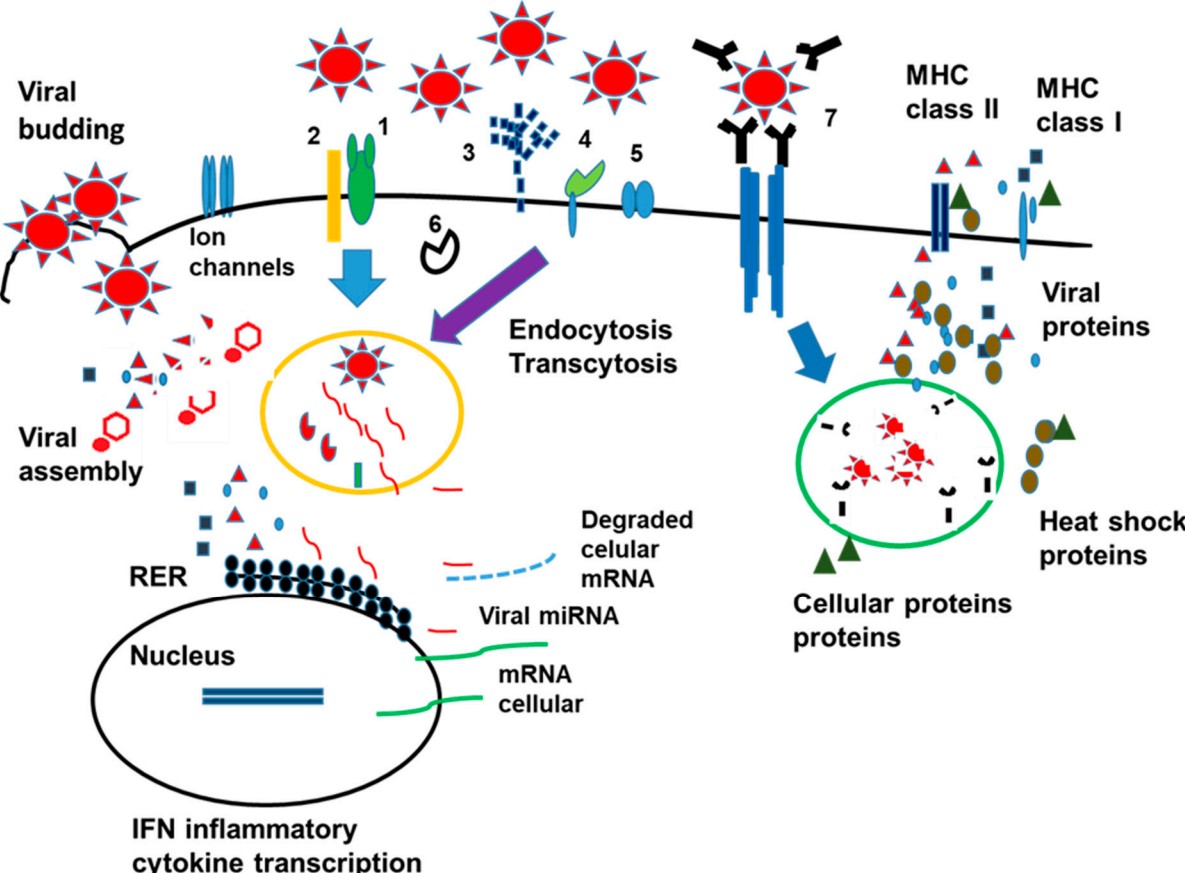

**Figure 1.** SARS-CoV-2 cell infection and primary cell response. The figure represents the different processes related to viral infection and cell response. The primary receptor is the ACE2 (1) receptor, along with the protein TMPRSS2 (2); heparan sulfate or heparin (3) is another receptor, especially in cells not expressing the ACE2 receptor; sintenin-1 and PALS1 proteins that interact with E protein are represented by number (4). The other type of receptors are CD26, CD147 and neuropilin 1 (5). The virus can also enter by transcytosis, binding to different proteins depending on the cell type (6). The complex virus antibodies are endocytosed through the Fc receptors, highly expressed in phagocytic cells (7). The virus can induce cellular mRNA degradation by miRNA. Viral proteins block IFN signal transduction, the transcription factor NFkB, which is crucial for the transcription of proinflammatory cytokines. Viral proteins can be presented as antigens alone or in conjunction with heat shock or other cellular proteins. Viral proteins can also be secreted into the media. Ion channels are essential for inflammasome activation but are also crucial in cell homeostasis.

Both antigens, CD26 and CD147, have been linked to autoimmune diseases [64–67]. Pharmacological inhibition of dipeptidyl dipeptidase has been studied in autoimmunity and COVID-19 [64–67]. Autoantibodies against CD26 have been described in naïve patients with rheumatoid arthritis [67]. More research is required to define a link between both antigens, COVID-19 and autoimmunity.

The infection of other cells, like the senescent endothelial cells, can facilitate virus replication and immune cell activation [68]. Endothelial dysfunction as a product of viral infection may be responsible for vascular events, primarily seen in patients with comorbidities [68,69]. The presence of heparan sulfate or proteoglycans containing sugar moieties facilitates virus binding to the cell surface independently of ACE2; the process is related to more than one viral protein [38,69].

López-Muñoz et al. [70] demonstrated that viral N protein is present in T lymphocytes and is bound to heparan sulphate or heparin. Protein N on cell membranes modulates innate and adaptive immune responses [70]. It interacts with Smad3 leading to chloride accumulation in respiratory epithelium and activation of proinflammatory pathways [71]. A multicentric study demonstrated circulating N protein in patients infected with SARS-CoV-2 [72]. It may be inferred that this circulating protein is responsible for the continuous stimulation of the inflammatory process, especially for those patients with comorbidities. Currently, there are no clearly defined mechanisms in vivo to determine if the T cells are infected and then express the N protein or if the N protein is circulating and binding to receptors on T cells.

The viral envelope protein interacts with the PDZ domain of the membrane proteins sintenin-1 and PALS1, which activate receptors linked to cellular damage triggering the inflammasome and inducing the cytokine storm [73,74]. Interestingly, T lymphocyte responses in longitudinal studies are preferential to S and N viral proteins, not to the E protein [75]. These results suggest that the E protein may (1) contribute to either the inflammation induced by continuous viral replication without being blocked by antibodies, or (2) facilitate infection by acting as a co-ligand of viral infection using the heparin sulfate or heparin and sintein-1/PALS as coreceptors.

Rashid and coworkers [22] reviewed the importance of NSP, ORF and M proteins in SARS-CoV-2 infection. Different cell targets bind NSP, ORF and N viral proteins. The NSP1 protein binds to the 40S ribosomal unit of the host's ribosome, blocking protein transcription. Since IFN transcription is induced as an initial alert, inhibition of the 40S ribosomal unit affects the transcription of IFNs and favors viral exocytosis [21]. NSP2 interacts with a family of prohibiting proteins that play a role in mitochondrial metabolism, impairing proper T cell stimulation and activation [21]. The signal transduction protein IRF3 can be inhibited by NSP1, NSP6, NSP8, NSP12, NSP14 and NSP15 [22]. NSP1, NSP6 and NSP13 inhibit STAT1 [22]. In addition, the proteins NSP1, NSP2 and M inhibit the activation of the transcription factor NFkB.

ORF3a, ORF6, ORF7a, ORF8, ORF9b and ORF10 affect the interaction between the host and virus [21–23]. ORF3a affects the apoptosis and autophagy pathways leading to cellular inactivity; ORF3a and ORF6 inhibit STAT1 signaling [21,22]. ORF7b blocks STAT1 and STAT2 activation. On the other hand, ORF7a activates NFkB and thus the transcription of proinflammatory cytokines, but it blocks STAT2, which plays a role in IFNs signaling pathways [21,22]. ORF8 blocks antigen presentation by stopping antigenic expression [21,22]. ORF9b blocks the mitochondrial-induced antiviral effect utilizing heat shock proteins (HSP) [21,22]. Finally, ORF10 blocks mitochondrial antiviral protein, inducing viral replication [22].

Viral proteins are essential to induce an antiviral immune response; however, as described, viral proteins interact with cellular proteins impairing the immune response and could be presented as antigens [61]. Thus, high virus titers or constant replication lead to more viral protein secretion, activating the inflammasome and provoking a cytokine storm, leading to viral escape.

There are different classes of antibodies targeting all structural proteins of the virus. Neutralizing antibodies are of utmost importance, given their ability to bind to specific sites of the S protein that bind to the receptor (RGD region) [76,77]. The most important one is the one that blocks viral fusion with the cell membrane of respiratory epithelium in the presence of angiotensin 2 converting enzyme by acting as a receptor for the virus. The antibody response in some patients is not protective and could cause even more damage (illness mediated or worsened by antibodies) [76,77]. Autoantibodies against CD26 or other receptors could be present in severe or prolonged COVID-19 individuals, as shown in other diseases [67]. The production of polyclonal antibodies does not protect from the viral infection, and neutralizing antibodies block only part of the interaction of the virus with target cells. Thus, individuals can still be infected, but the virus infection can be controlled more efficiently.

Innate immune response evasion during the first phase of infection (nasal phase) determines viral replication leading to viral pneumonia and viral sepsis. Patient recovery is also affected by viral replication. An overstimulated adaptive immune response—uncontrolled cytokine production, especially IL-6, dysfunctional complement and coagulation cascades—leads to disease progression and severity [38,78–84]. Some variants of SARS-CoV-2 cause increased interferon resistance, promoting innate immune response evasion [85]. Thus, an ineffective and exacerbated immune response can be responsible for multi-organ damage in COVID-19 patients.

A summary of the text, viral infection, cellular compartments, viral protein secretion and presentation as antigens is depicted in Figure 1.

### 3. Autoimmunity and COVID-19

COVID-19 patients produce a broad spectrum of antibodies due to molecular mimicry, especially the sequences of the most mutated protein, S [38,61]. Molecular mimicry and cell stress induced by viral infection partially explain the formation of antibodies against most immunomodulatory proteins, cytokines, chemokines, complement, and some cell intracellular and surface proteins [61,86] (Table 1). This autoreactivity can lead to thrombi formation and neurological events. Blocking immune receptors or cytokines impairs the immune response against the virus leading to viral escape [86–88]. The disruption of immune tolerance, the increased cell death induced by the non-apoptotic elimination of virus-infected cells, the uncontrolled secretion of cytokines (TNFα, IFNβ, IL-6, IL-1β; IL-17 e IL-18) and the abnormal expression of antigens upon virus infection and inflammation are responsible for the generation of autoantibodies.

**Table 1.** The most prevalent autoantibodies described during a SARS-CoV-2 infection and in prolonged COVID-19 [86–130].

| Wide Spectrum Antibodies | Specific Antibodies |
|---|---|
| <ul><li>Antinuclear antibodies</li><li>Anti-phospholipids (anti β2 glicoprotein1, anticardiolipin, antiphosphatidylserine/prothrombin)</li><li>Anti-cytoplasmic antibodies (ANCA)</li><li>Anti-platelet antibodies</li><li>Anti-Annexin V</li><li>Anti-glia antibodies</li><li>Anti-neuron antibodies</li></ul> | <ul><li>Anti-cytokines: IFN α, IFN ω, IFNγ, IL1β, IL-6, IL-10, IL-17, IL-21, GMCSF</li><li>Anti-chemokines: CCL2, CXCL1, CXCL7, CXC L13, CXCL16</li><li>Anti-complement proteins</li><li>Anti-ACTH</li><li>Anti-thyroglobulin</li><li>Anti-glutamic acid decarboxylase</li></ul> |

Wang and coworkers [86] described an increased proportion of antibodies against immunomodulatory proteins (cytokines, chemokines, complement and cell surface proteins) in 194 patients infected with SARS-CoV-2 compared to non-infected individuals. The authors also described how autoantibodies inhibit immune cell activation, block immune cell response, and induce viral escape. They also tested the hypothesis in a murine model infected with SARS-CoV-2 [86].

Antinuclear antibodies were reported in a small sample of patients with COVID-19-associated respiratory failure [87]. The cytoplasmic pattern was the most prevalent (64%), and its presence was linked to the worst clinical outcomes and prognosis [87]. Pascolini et al. [88] recorded that out of 33 patients: 45% were positive for at least one antibody, 33% were positive for antinuclear antibodies, especially nucleolar or spotted pattern, 24% were positive for anticardiolipin, and 8% were positive for anti-β2-glycoprotein I. None of those patients was positive for neutrophil anti-cytoplasmic antibodies; however, several groups have reported them [88–90]. Zuo et al. [91] reported the presence of antiphospholipid antibodies (anticardiolipin, anti-β2-glycoprotein I, anti-phosphatidylserine/prothrombin) in 52% of 172 hospitalized COVID-19 patients. Those antibodies could contribute to thrombi formation in some patients and neutrophil hyperreactivity, including the release of

NETs (neutrophil extracellular traps), mast cell activation, increased platelet aggregation, augmented respiratory syndrome severity and lower glomerular filtration rates [92,93]. Antiplatelet and procoagulant antibodies and autoimmune thrombocytopenia have been reported in vaccinated or infected individuals [93–96].

Seeßle et al. [97] reported that 43.6% of prolonged COVID-19 patients (post-acute COVID-19 sequelae) were positive for antinuclear antibodies ($\geq 1/160$) after a year, most of them women. High levels of autoantibodies in patients with prolonged COVID-19 are strongly linked to neurocognitive symptoms [97]. Autoantibodies were related to the common signs of fatigue, myalgia and breathing difficulties during low-intensity exercise [97].

Patients with severe COVID-19 have low levels of IFN type I due to the presence of anti-IFN antibodies [98–106]. Up to 10% of elderly patients are positive for this autoantibody before viral infection, which may increase with severity [98–106]. The low detectable levels of IFN type I in human samples are associated with increased dissemination of viral particles and pulmonary and systemic inflammation. Rather than a low production by the cells, the decrease in IFN type I and type III may be due to the presence of autoantibodies, which may be responsible for this effect [98–106]. Bastard et al. [100,101] demonstrated that ant-interferon autoantibodies were most frequent in men (94%) and elderly patients and that these were present before being infected with SARS-CoV-2 [100–105]. Children with autoimmune polyglandular syndrome have type I anti-IFN $\alpha 2$ antibodies, which explains the risk of developing COVID-19-induced pneumonia [106,107]. On the other hand, SARS-CoV-2 causes a temporary increase in preexisting anti-interferon antibodies [105–107]. This effect may be due to the overstimulation of B cells [38,39].

Neurological symptoms caused by COVID-19 have also been associated with anti-neuron crossed antibodies [108]. Franke et al. [109] demonstrated the presence of autoantibodies in critically ill COVID-19 patients. The patients presented neurological symptoms with unexplainable etiologies. The authors found elevated levels of anti-neuron and anti-glia autoantibodies in cerebrospinal fluid. However, these autoantibodies lack specific immunofluorescent patterns [109], suggesting the lack of specific proteins as targets.

Among the alterations reported in SARS-CoV-2 infection, patients with severe COVID-19 experienced hyperglycemia [110]; in most patients, normal blood sugar levels returned to normal upon recovery [110]. On the other hand, a group of post-COVID-19 patients developed diabetes mellitus type I with high titers of anti-glutamic acid decarboxylase antibodies (anti-GAD65) [111]. It is unknown if these autoantibodies will persist in time. Hypoxic encephalopathy and neuro-hypophysitis, the product of immune damage, have been proposed to explain the diabetes insipidus recorded in patients that suffered severe respiratory failure during COVID-19 infection [96,97].

There have been reports of autoantibodies linked to ACTH and thyroid hormone. Some SARS-CoV-2 and SARS-CoV amino acid sequences are homologous with the adrenocorticotrophic hormone (ACTH) [112–114]. Researchers believe that these similarities lead to the formation of cross-reactive antibodies responsible for the inactivation or destruction of endogenous ACTH [114–116]. These autoantibodies may be accountable for central hypocortisolism after viral infection. This event results in the evasion of the cortisol-mediated stress response [114–118].

Autoimmune thyroiditis has also been reported as a consequence of COVID-19. It is likely associated with molecular mimicking, polyclonal activation of T cells by antigenic epitopes related to the virus and an increment of antigens in thyroid cells due to augmented human HLA expression [115,119–122]. It is unclear if viral replication in the oromucosal area generates inflammatory responses that may induce the expression of thyroid antigens in patients with a predisposition.

B cell activation is also crucial in the autoimmune response observed in COVID-19. Woodruff et al. [122] described how naive B cells start producing autoantibodies in patients with severe SARS-CoV-2. Autoantibody levels decrease with time in most patients, but the question is whether the virus or the viral proteins are responsible for the increase in autoimmune responses. Castleman and coworkers [123] refer that autoantibodies are linked

to the alteration of double-negative B cells, suggesting a lack of regulation and tolerance. One plausible hypothesis is that the membrane expression of viral proteins is responsible for maintaining T and B cell stimulation. Immune complexes can also be accountable for chronicity in prolonged COVID-19 [38,62].

Other autoimmune conditions, such as idiopathic thrombocytopenia [124], systemic lupus erythematosus [125] and autoimmune neurological illnesses, multiple sclerosis [126,127], optical neuromyelitis [128], Guillain-Barré [129] and myasthenia gravis [130], have also been reported to occur or flare in post-SARS-CoV-2 infection.

The risk of coinfection with opportunistic pathogens is very high in patients with autoimmune diseases or with autoantibodies against cytokines [131]. Two critical events relate to the mimicry between pathogens, cell proteins, and lipids. One, antigen T cell hyperactivation in autoimmune responses may drive these cells to exhaustion and anergy [132–134]. Two, the increase in autoantibodies against IFNs slows the immune response against the virus. The direct or indirect activation of the inflammasome and cells generate a sepsis-like multi-organ failure [132–134]. More studies are needed to determine the risk of autoimmunity after viral infection.

## 4. Primary Immunodeficiency and COVID-19

Immunodeficiencies are strongly linked to severe COVID-19 [135–141]. Patients with primary or secondary immunodeficiency require more time to eliminate the virus [137–141]. However, in patients with primary immunodeficiency, the exacerbation of the immune response is less probable. Only patients with comorbidities or demographic factors are more susceptible to developing severe COVID-19 disease [135–146].

In primary immunodeficiency patients, mutations in proteins involved in IFN type I and IFN type III transcription and signal transduction often progress to severe disease [145]. Patients with combined immune defects may suffer chronic COVID-19 infection [134–136]. However, severe COVID-19 is rarely observed in children with primary immunodeficiencies involving antibody synthesis or phagocytosis [137–139]. These patients could be asymptomatic carriers of the virus [137]. It is unclear if normal children develop long-lasting memory immune responses against the virus [140].

Patients with Common Variable Immunodeficiency (CVI), the most common primary immunodeficiency among children and adults, usually have severe infections, high inflammatory sequelae, and a high incidence of autoimmune diseases and cancer [135,141]. The condition involves multiple genetic defects [135,141]. These patients typically develop severe COVID-19, especially when the number of circulating T lymphocytes is low [141]. On the contrary, patients with only minor antibody deficiencies develop mild COVID-19, probably due to a decreased inflammatory response generated by the immune complex in severe disease [11,80,81,83,139–141]. Patients with chromosome X-associated agammaglobulinemia (Bruton's agammaglobulinemia) with no comorbidities experienced only mild COVID-19 [142]. Pulverenti and coworkers [143] reported an excellent response to the vaccine and a mild disease course in their cohort of patients with 22q11.2 deletion syndrome [36,143]. It is assumed that the benefit in these patients is the lack of pathologic antibodies or autoantibodies that usually impede the resolution of inflammation induced by the virus. Intravenous immunoglobulins have been helpful in CVI with low lymphocyte count and comorbidities [36,144].

T cell deficiencies, especially CD4+ T cells, HIV infection, immunosuppressants or chemotherapy, are associated with severe COVID-19 and a higher risk of ICU admission [36–141,145]. Patients treated with anti-CD20, anti-CD19 or CAR T-Cell (chimeric antigen receptor of T cells) and those with B cell deficiencies, hypogammaglobulinemia and altered T cell number or function have a higher risk for developing severe COVID-19, uncontrolled viral replication and slower viral elimination [146–148]. However, as described in CVI, treatment with monoclonal antibodies or intravenous immunoglobulins has been helpful in these patients [36,144].

Patients with genetic defects affecting Type I IFN amplification and/or induction with low or nonexistent levels of IFNs, also develop severe COVID-19 [98–105]. Mutations in the genes involved in IFN type I signaling, occurring only in 1–5% of healthy young populations, are also linked to severe COVID-19 [147]. Mutations on IRF3, IRF7, IRF9, and IFNAR2 signaling proteins are related to the lack of IFN types I–III signaling, leading to immune evasion [147,148]. As mentioned previously, viral proteins can inhibit IFN signaling and IRF3 [21,22].

Approximately 3% of adults with severe COVID-19-induced pneumonia have innate autosomic TLR3 defects (despite a typical IFN type I response). These defects include recessive autosomic forms of IFNAR1 and IRF7 described in healthy adults [147,148]. Chromosome X-linked TLR7 deficiency is present in 1% of the male population. It is linked to severe pneumonia caused by the virus and could even develop telangiectasia ataxia [149,150]. Other IFN-I-mediated innate impairments, STAT2 or TYK2 deficiencies, are linked severe illness in children and adults [36,102–104,147,148]. As described before, viral proteins can inhibit STAT1, STAT2 and NFkB activation, enhancing the lack of immune response [21,22].

In a recent study, conducted in the Czech Republic by Milota and coworkers [151], there was a 2.3-fold increase in hospitalizations and increased mortality (2.4% vs. 1.7% of the general population). In these patients, COVID-19 severity was associated with lymphopenia and hypogammaglobulinemia, not with age or BMI [151]. Individuals with hereditary angioedema don't develop severe disease despite having an altered bradykinin metabolism [129]. Treatment with monoclonal antibodies against the S protein and the administration of plasma from recovered patients enhanced the antiviral response [151]. However, monoclonal antibody therapies are ineffective against the new variants and subvariants of the SARS-CoV-2 virus [152–154]. The results of a UK cohort that included 117 primary immune deficient patients were similar to the Czech study; only 22.7% of the patients had the infection, and 85% had a mild course of the disease [152].

The effects of the vaccines against SARS-CoV-2 will depend on the type of immunodeficiency. The impact of the RNA and vector vaccines is likely similar in patients with antibody immunodeficiency and just as effective as tetanus or the HiB vaccine [81,143,154].

Anti-cytokine antibodies are associated with immunodeficiency, especially in the case of anti-IFN antibodies ($\alpha$, $\gamma$, $\omega$), IL-6, IL-17 (A, F), [98–105,155]. Anti-IFN $\gamma$ antibodies block cell signaling dependent on the cytokine, the membrane expression of HLA-DR, the secretion of TNF$\alpha$ and IL-12 and the transcription of genes induced by IFN $\gamma$ [156–158]. Anti-cytokine antibodies (especially anti-IFN-$\alpha$) increase with age and are unrelated to COVID-19 infection [98–105,155–158]. Anti-IFN $\alpha$ and anti-IFN $\omega$ autoantibodies are observed in 13% of critically ill COVID-19 patients [98–105], and this percentage increases to 21% in patients older than 80 infected with the virus [98]. Anti-IFN type I [$\alpha$ and $\omega$] autoantibodies were found in 24% of the patients who developed severe COVID-19, despite being vaccinated [101], and up to 18% of the patients that die after a COVID-19 infection have anti-IFN type I antibodies [101].

Anti-cytokine antibodies have been described in several medical conditions involving autoimmune and infectious diseases [155,156]. The presence of anti-cytokine antibodies in several infectious diseases, which induces a secondary immune deficiency, has been termed "phenocopy" of primary deficiencies [156]. The term "phenocopy" refers to a clinical phenotype related to the abnormal genetic variants involved in cytokine pathways. The higher titer of autoantibodies affects the cellular response, masking the effect induced by the pathogen [155–157]. Thus, anti-cytokine autoantibodies should be screened in healthy elders and high-risk individuals since their detection may orient physicians to use effective therapies to prevent and treat severe and chronic diseases associated with infections.

Critically ill COVID-19 individuals had a reduced IFN type I response and increased expression of inhibitory receptor LAIR-1 (leukocyte-associated immunoglobulin receptor 1) in monocytes (19%) despite the presence or absence of anti-IFN antibodies [158]. These

results support the role of inhibitory receptors in the impaired immune response and the reduction of biologically active IFN during COVID-19 [103,159].

Recently, Casanova and Abel reviewed the complex interactions between genetic mutations, autoimmunity, and the immune response against pathogens [160]. There is still room for learning the complex interaction between immunodeficiency, autoimmune-induced immune deficiency, and genetic alterations unrelated to immune cells.

## 5. Conclusions

It is crucial to test for primary and secondary immunodeficiencies in patients with severe and prolonged COVID-19. Subtle genetic mutations should be screened in young individuals developing moderate and severe COVID-19. A group of secondary immunodeficiencies are associated with anti-immunoregulatory antibodies against anti-IFN I [$\alpha$ y $\omega$] or other cytokines that could be easily detected. This autoantibody screening can assist the physician in analyzing previously unknown monogenic or oligogenic causes of autoimmune and immune deficiency diseases. Primary immunodeficiency, secondary immune deficiency, and autoimmune diseases can be treated accordingly.

SARS-CoV-2 and other viruses induce a polyclonal activation of B cells. This polyclonal activation, which may be transient, results in increased production of autoantibodies. The lack of tolerance and probably an impaired T cell thymic education affects the production of autoantibodies.

Figure 2 summarizes the interaction between immunodeficiency and autoimmunity in SARS-CoV-2 infection and prolonged COVID-19. In the graph, the variety of autoantibodies responsible for the impaired immune response against pathogens are more significant than the genetic mutations responsible for the lack of immune response. As described, patients with autoantibodies against cytokines have a higher incidence of severe disease.

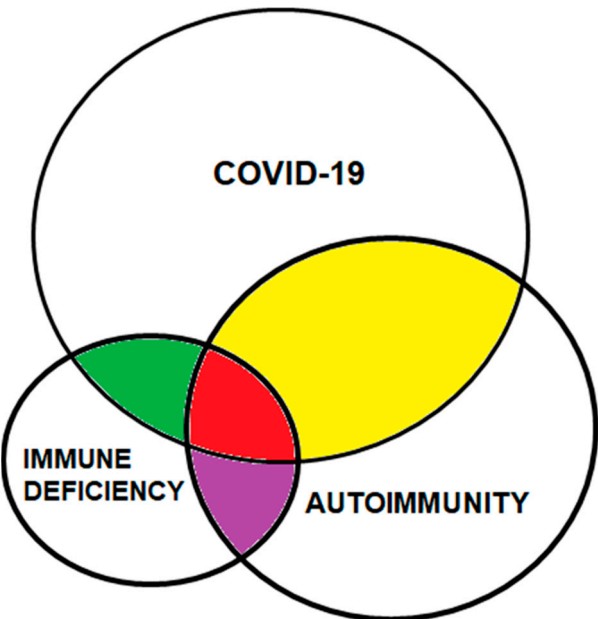

**Figure 2.** Involvement of autoimmunity and immunodeficiency in COVID-19. Autoimmunity, unlike genetic mutations related to immunodeficiency, is frequent in severe and prolonged COVID-19. The yellow segment corresponds to antibodies against essential proteins within the immune response against COVID-19. The red area represents the ratio of autoantibodies causing immunodeficiencies (antibodies against cytokines, for example). The green area corresponds to genetic mutations related to an inadequate response against the virus. The purple segment represents the relationship between autoimmunity and immunodeficiency independent of COVID-19.

Patients with unknown or uncontrolled metabolic or neurodegenerative disorders may have a higher risk of severe or prolonged COVID-19. In addition, the associated risk should be screened depending on gender; some autoimmune manifestations are more prevalent depending on gender. Even though patients with chronic inflammatory diseases should be more susceptible to severe or prolonged COVID-19, only those without properly controlled therapy were shown to have a high risk of developing severe or prolonged COVID-19 [161,162].

The impact of comorbidities and pharmacological treatment should be carefully analyzed. There are mixed opinions over the protective effects of metformin and oral steroids in patients with various conditions [163–165]. Interesting approaches have been published regarding score models to standardize a prediction of hyperinflammation in COVID-19 [166]. However, more research on severe and prolonged COVID-19 individuals is needed to decrease the morbidity and mortality of this disease. The lessons learned from COVID-19 disease can be helpful for subsequent viral outbreaks.

## 6. Limitations

The review aims to highlight autoimmunity-induced immunodeficiency's importance in the SARS-CoV-2 infection and COVID-19 in healthy individuals. However, autoantibody screening, especially for anti-cytokines, is not a standard parameter measured in preventive medicine. The other limitation is the heterogeneity of the screening, the antibodies analyzed, and the time period. Well-designed clinical trials are needed. Despite more than two years of the pandemic, we are still learning about SARS-CoV-2, COVID-19 and prolonged COVID-19.

**Author Contributions:** Conceptualization, J.V.G., C.V.D.S. and A.H.G.; writing—original draft preparation, J.V.G., C.V.D.S. and A.H.G.; writing—review and editing, J.B.D.S. and M.H.; funding acquisition, M.H. All authors have read and agreed to the published version of the manuscript.

**Funding:** The authors were financed by the following projects: [1] Multicenter Project funded by the Czech Ministry of Innovation FW03010472 Study of the efficacy of experimental vaccines against SARS-CoV-2 in animal models (JBDS, MH); [2] a grant from the European Structural and Investment Operational Funds Program Research entitled: Molecular, cellular, and clinical approach to healthy ageing grant ENOCH; Registration number: CZ.02.1.01/0.0/0.0/16_019/0000868, project lieder at the IMTM, Palacky U, MH; [3] National Institute of virology and bacteriology [Program EXCELES, ID Project No. LX22NPO5103]—Funded by the European Union (JBDS, JVG); [4] The Venezuelan Foundation of Scientific Research and Technology (FONACIT) and the Ministry of Science, Venezuela, grants to the Institute of Immunology, Faculty of Medicine, Universidad Central de Venezuela, responsible for Dr Alexis García.

**Institutional Review Board Statement:** Not applicable.

**Informed Consent Statement:** Not applicable.

**Acknowledgments:** In memoriam of Nicolás Enrique Bianco Colmenares (1943–2022).

**Conflicts of Interest:** The authors declare no conflict of interest.

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
