# Peer review of "Autoimmunity and Immunodeficiency in Severe SARS-CoV-2 Infection and Prolonged COVID-19"

_cimb, doi:10.3390/cimb45010003_

Round 1

Reviewer 1 Report

This manuscript proposes to review the links of COVID-19 and SARS-CoV2 to autoimmunity and immune deficiencies (Title: Autoimmunity and immunodeficiency in severe SARS-CoV-2 infection and prolonged COVID-19). The review is generally interesting and clearly written.  However, this reviewer believes the following comments and concerns need to be addressed.

1.       Unfortunately, this is not a systematic review (or meta analysis) and many publications that have alternative views on some of the claimed links of COVID-19 to autoimmunity and immune deficiency were not cited.

2.       Approximately 1/3 of the text at the beginning of the manuscript is devoted to the molecular biology and pathogenesis of SARS-CoV2. This leaves the reader wondering how this section is related to the title and the presumed focus on autoimmunity and immune deficiencies. Much of this part of the manuscript has already been published by at least some of the co-authors (reference 31). Figure 1 is interesting and nicely presented, but how is it linked to autoimmunity or immune deficiencies? MHC is a part of this figure but is not mentioned as a factor in autoimmunity or immune deficiencies. Perhaps the intention was to address this in one sentence (lines 379-380)?

3.       The section on autoimmunity and immune deficiencies in the setting of COVID-19 lacks balance. There is no critical appraisal of many of the publications cited (they are seemingly taken at face value). For example, many publications cited relied on comparisons to healthy individuals as controls begging the question as to how the observations are specific for SARS-CoV2 (see reference 71 and DOI: 10.1136/annrheumdis-2021-220520; and others). As another example, refence to “cytokine storm” (lines 105 + 143) might be balanced by clarifying that this ‘cytokine storm’ is unlike that seen prior to COVID-19 (levels of many cytokines often do not approach those seen in Macrophage Activation Syndrome. see: doi: 10.1002/jmv.26317 for one example).

4.       Line 82: to be correct, it should probably be stated that the abrogation and elimination of SARS-CoV2 requires effective immune responses.

5.       Lines 92-95; line 116; lines 206-207; lines 309-310. Please provide references for these statements.

6.       Lines 151-152: “40S region” and “40S” should be ‘the 40S ribosomal subunit’.

7.       Table 1: why are specific anti-phospholipid antibodies regarded as “wide spectrum”. And “Anti-cytoplasmic antibodies (ANCA)” should be anti-neutrophil cytoplasmic antibodies (ANCA)’. Anti-platelet should be hyphenated. Unfortunately, the list is far from comprehensive and it is not clear how those in this table were adjudged to be the “most prevalent”? How was that calculated or determined? Correct spelling of prolonged in title.

8.       Lines 372-373: it is not clear why antibodies to these interferons “should be detected in elders and high risk individuals”. What would be actionable if they were positive?

9.       The manuscript needs at least a paragraph pointing to the limitations of this review (some referred to above (i.e., not a systematic review). There is virtually no mention of multi-inflammatory syndrome in children. There is also no mention that there was little apparent significant increased frequency of severe COVID-19 in many systemic autoimmune diseases (i.e., SLE).

10.   Unfortunately, figure 2 adds very little to the manuscript. Are the sizes of the circles in this Venn diagram significant? It is stated that the red area represents a “ratio”…how was the ratio derived? Correct spelling of Autoimmunity.

11.   There are a number of other spelling and grammatical errors (too numerous to cite completely here). A partial list:

a.       Line 38: “...non-segmented positive RNA and a characteristic protein in…” should be ‘non-segmented positive-stranded RNA’.

b.       Line 41: spell out R0 first time.

c.       Line 49: “especially epithelial cells type II and club cells with lesser intensity..”. do you mean that it binds both type II epithelial cells and club cells with lesser intensity, or only club cells with lesser intensity?

d.       Line 65 and numerous instances throughout the manuscript (lines 210; 229; 276) words are inappropriately hyphenated. Here ve-nous should not be hyphenated.

e.       Line 75: “transplanted patients” is a colloquialism and should be corrected. Patients who have received an organ or tissue transplant?

f.        Line 89 and elsewhere (line 108): Phagocyte and endocyte (subcellular organelles or processes) should be ‘phagocytosis’ and ‘endocytosis’.

g.       Line 96 and elsewhere. is IFN I should be Type 1 IFN?

h.       Line 221: “mice” should be ‘mouse’ or ’murine’

i.         Line 241: “are” should be ‘were’

j.         Line 271: “crossed antibodies” should be cross-reactive antibodies’?

k.       Line 276: “molecular mimicking’ should be ‘molecular mimicry’.

l.         Line 291: “occur or aggravate” should be ‘occur or flare’?

m.     Lines 356-358 should be combined into one sentence.

Author Response

REVIEWER 1.

This manuscript proposes to review the links of COVID-19 and SARS-CoV2 to autoimmunity and immune deficiencies (Title: Autoimmunity and immunodeficiency in severe SARS-CoV-2 infection and prolonged COVID-19). The review is generally interesting and clearly written.  However, this reviewer believes the following comments and concerns need to be addressed.

We would like to thank the reviewer for his dedication in reviewing the manuscript and provide specific questions and correcting parts of the text that were either unclear or there were typo or grammatical mistakes. The text was modified as well as the figures and several references have been added.

  1. Unfortunately, this is not a systematic review (or meta analysis) and many publications that have alternative views on some of the claimed links of COVID-19 to autoimmunity and immune deficiency were not cited.

The aim of the article is to review the current view of autoimmunity and primary immunodeficiency in severe and prolonged COVID-19 in previously "healthy " individuals. We could not find original articles that support alternative views on probable mechanism of immune response impairment aside from primary immune and autoimmunity.

  1. Approximately 1/3 of the text at the beginning of the manuscript is devoted to the molecular biology and pathogenesis of SARS-CoV2. This leaves the reader wondering how this section is related to the title and the presumed focus on autoimmunity and immune deficiencies. Much of this part of the manuscript has already been published by at least some of the co-authors (reference 31). Figure 1 is interesting and nicely presented, but how is it linked to autoimmunity or immune deficiencies? MHC is a part of this figure but is not mentioned as a factor in autoimmunity or immune deficiencies. Perhaps the intention was to address this in one sentence (lines 379-380)?

The text of the section was modified accordingly as well as the figure

  1. The section on autoimmunity and immune deficiencies in the setting of COVID-19 lacks balance. There is no critical appraisal of many of the publications cited (they are seemingly taken at face value). For example, many publications cited relied on comparisons to healthy individuals as controls begging the question as to how the observations are specific for SARS-CoV2 (see reference 71 and DOI: 10.1136/annrheumdis-2021-220520; and others). As another example, refence to “cytokine storm” (lines 105 + 143) might be balanced by clarifying that this ‘cytokine storm’ is unlike that seen prior to COVID-19 (levels of many cytokines often do not approach those seen in Macrophage Activation Syndrome. see: doi: 10.1002/jmv.26317 for one example).

We have modified the text of the manuscript accordingly.

  1. 4. Line 82: to be correct, it should probably be stated that the abrogation and elimination of SARS-CoV2 requires effective immune responses.

We apologize for the mistake the text was corrected.  

  1. Lines 92-95; line 116; lines 206-207; lines 309-310. Please provide references for these statements.

The references were added.

  1. Lines 151-152: “40S region” and “40S” should be ‘the 40S ribosomal subunit’.

We apologize for the mistake

  1. Table 1: why are specific anti-phospholipid antibodies regarded as “wide spectrum”. And “Anti-cytoplasmic antibodies (ANCA)” should be anti-neutrophil cytoplasmic antibodies (ANCA)’. Anti-platelet should be hyphenated. Unfortunately, the list is far from comprehensive and it is not clear how those in this table were adjudged to be the “most prevalent”? How was that calculated or determined? Correct spelling of prolonged in title.

In several autoimmune disorders circulating ANA, ANCA, anti-phospholipid are observed and therefore are not considered specific. The polyclonal activation of B cells due to virus infection may be responsible for these autoantibodies as clarified in the text. The figure was corrected.

  1. 8. Lines 372-373: it is not clear why antibodies to these interferons “should be detected in elders and high risk individuals”. What would be actionable if they were positive?

Commercial ELISA kits for anti-IFN alfa2 are available from Thermo FIsher. A suggested range can be found in the protocol.

  1. The manuscript needs at least a paragraph pointing to the limitations of this review (some referred to above (i.e., not a systematic review). There is virtually no mention of multi-inflammatory syndrome in children. There is also no mention that there was little apparent significant increased frequency of severe COVID-19 in many systemic autoimmune diseases (i.e., SLE).

We introduced a text concerning limitations of the review.

  1. Unfortunately, figure 2 adds very little to the manuscript. Are the sizes of the circles in this Venn diagram significant? It is stated that the red area represents a “ratio”…how was the ratio derived? Correct spelling of Autoimmunity.

Figure 2 and the text was corrected.

  1. There are a number of other spelling and grammatical errors (too numerous to cite completely here).

We apologize for the mistake sand we would like to thank the reviewer for the corrections. The text was corrected

Reviewer 2 Report

The manuscript is an incomplete review. Much important information should be provided to update the references. Some sentences should be reformulated, in particular the sentences concerning patients with primary immunodeficiencies. For example, in CVID the course is not severe in the majority of patients. There is a lack of information on primary T lymphocyte defects such as chromosome 22 Deletion Syndrome. In essence, the manuscript requires a profound revision as every single sentence cannot remain generic because it is a review

Author Response

The manuscript is an incomplete review. Much important information should be provided to update the references. Some sentences should be reformulated, in particular the sentences concerning patients with primary immunodeficiencies. For example, in CVID the course is not severe in the majority of patients. There is a lack of information on primary T lymphocyte defects such as chromosome 22 Deletion Syndrome. In essence, the manuscript requires a profound revision as every single sentence cannot remain generic because it is a review

We would like to thank the reviewer for the valuable comments.  The review article was modified accordingly, references were added in the text and 20 new references were added.

Round 2

Reviewer 1 Report

Thank you for the responses.

Arguably, there have been no longitudinal studies of previously healthy individuals that developed COVID-19 and features of autoimmunity. The use of normal health individuals as controls does not address the question.

The authors misunderstood item 8: 

  1. it is not clear why antibodies to these interferons “should be detected in elders and high risk individuals”. What would be actionable if they were positive?

Commercial ELISA kits for anti-IFN alfa2 are available from Thermo FIsher. A suggested range can be found in the protocol.

This response does not explain WHY the antibodies should be detected in elderly and at risk individuals. It merely indicates HOW they can be detected.  If antibodies to interferons are detected what would be the action that would follow? How would a physician use this information?

Author Response

We would like to thank the reviewer for the valuable comments and apologize for the mistake. The manuscript has been modified to answer the question. Two references have been added regarding the clinical relevance of detecting autoantibodies against cytokines and the importance of their screening in monogenic or polygenic primary immune deficiency and autoimmunity. 

The text was corrected after the revision. All the changes are highlighted in yellow.